



# Assessing human-caused wildfire ignition likelihood across Europe

Pere Joan Gelabert[a], Adrián Jiménez-Ruano[a,b,c], Clara Ochoa[d], Fermín Alcasena[e], Johan Sjöström[f], Christopher Marrs[g], Luís Mário Ribeiro[h], Palaiologos Palaiologou[i], Carmen Bentué Martínez[j], Emilio Chuvieco[d], Cristina Vega-García[a], Marcos Rodrigues[a,b,c*]

[a] *Department of Agriculture and Forest Engineering, University of Lleida, Alcalde Rovira Roure 191, 25198 Lleida, Spain*

[b] *GEOFOREST Group, University Institute of Research in Environmental Sciences (IUCA), University of Zaragoza, Zaragoza, Spain*

[c] *University Institute of Research in Environmental Sciences (IUCA), University of Zaragoza, Zaragoza, Spain*

[d] *Department of Geography, University of Alcalá de Henares, Alcalá de Henares, Madrid, Spain.*

[e]*Institute for Sustainability & Food Chain Innovation, Department of Engineering, Public University of Navarre, Campus Arrosadia, 31006 Pamplona, Spain*

[f] *Research Institutes of Sweden (RISE), Göteborg, Sweden*

[g] *Technische Universität Dresden, Dresden, Germany*

[h] *Department of Mechanical Engineering, Universidade de Coimbra, ADI, Coimbra, Portugal*

[i] *Department of Forestry and Natural Resources Management, Agricultural University of Athens, Karpenisi, Greece*

[j] *GEOT Group, University Institute of Research in Environmental Sciences (IUCA), University of Zaragoza, Zaragoza, Spain*

[*]*Corresponding to: Marcos Rodrigues rmarcos@unizar.es*

**Abstract.** This study features a cohesive modelling approach of human-caused wildfire ignitions applied to a set of representative regions in terms of fire activity across Europe (pilot sites, PS). Our main goal was to develop a common approach to model human-caused ignition probability at a fine-grained spatial resolution (100 m) and identify the main drivers of their emerge. Specifically, we (i) ascertain which factors influence ignitions in each PS; (ii) deliver a spatial-explicit representation of ignition probability, and (iii) provide a framework for comparison with regional-scale models among PS. To 25 do so, we calibrated Random Forest models from historical fire records compiled by local fire agencies, and geospatial layers of land cover, accessibility, population density and dead fine-fuel moisture content (DFMC). Models were built individually for each PS, comparing them with a full model constructed from all PS. Furthermore, special attention was given to the effect of spatial autocorrelation in model performance. All models achieved sufficient predictive performance (AUCs from 0.70 to 0.89). For all PS models, the yearly anomaly in DFMC was the most influential variable. Among human-related factors, 30 distance to the Wildland Urban Interface emerged as the most relevant variable, followed by proximity to roads, population density, and the fraction of wildland coverage. The performance of the full model achieved an AUC value of 0.81, with mean DFMC and anomaly being the main ignition factors, modulated by distance to roads and population density. The local performance of the full model dropped by 0.10 for AUC in both Southern Sweden and Attica (Greece) regions. The wildfire



occurrence models developed in this study are essential for understanding wildfire ignition hazard and may help implement

integrated wildfire risk management strategies and mitigation policies in fire-prone EU landscapes.

**Short summary:** Wildfires threaten ecosystems and communities across Europe. Our study developed models to predict where and why these ignitions occur in different European environments. We found that weather anomalies and human factors, like proximity to urban areas and roads, are key drivers. Using Machine Learning our models achieved strong predictive accuracy.

These insights help design better wildfire prevention strategies, ensuring safer landscapes and communities as fire risks grow with climate change.

## 1 Introduction

Wildfires are a common hazard that affect natural environments and human communities, especially in Mediterranean regions where fire risk is expected to increase in the forthcoming decades, linked to land abandonment, urban expansion on wildlands

and climate change (Colantoni et al., 2020; Dupuy et al., 2020; Salis et al., 2022). Knowing when, where and how the fires are going to start is crucial to guide integral fire management towards risk reduction. Wildfire ignition is a key component of wildfire risk assessment frameworks (Chuvieco et al., 2023), with human-caused fires playing a leading role in the human-dominated landscapes of Europe (Camia et al., 2013). The tendency of humans to initiate wildfires, especially during the high fire risk season when they can evolve to extreme wildfire events, has drawn considerable attention, research and funding over

the years, with manifold approaches emerging as fire information increased, and modelling techniques and processing capacity improved (Costafreda-Aumedes et al., 2017).

Several human-related drivers of wildfires have been identified. The prevalent role of the wildland-urban interface (WUI) as one of the main fire triggering spatial agent, has been extensively documented in several studies and regions across the world (Badia et al., 2011; Bar-Massada et al., 2023; Calviño-Cancela et al., 2017; Chuvieco et al., 2023; Schug et al., 2023; Syphard

et al., 2007). On the other hand, agricultural activities are also a noteworthy source of ignition, followed by a large fraction of negligent and accidental fires linked to different land management practices (e.g., use of machinery or disposing of residues), especially in Central-Northern of Portugal (Meira Castro et al., 2020) and Spain (Martín et al., 2018; Rodrigues et al., 2014; Tedim et al., 2022). Accessibility is also a key factor fostering ignition likelihood in proximity to roads and pathways (Chicas and Østergaard Nielsen, 2022; Costafreda-Aumedes et al., 2017; Oliveira et al., 2012).

In Europe, a multifaceted socio-economic trend has been driving the abandonment of traditional land activities (Lasanta and Vicente-Serrano, 2012). Land abandonment primarily results in woodland encroachment in formerly non-forested areas (Gelabert et al., 2021), which enhances the continuity and increases the availability of burnable material, among other effects. From a socio-demographic perspective, the massive immigration from rural areas to urban regions led to the abandonment of agricultural lands, thereby increasing the population density in metropolitan and agriculturally intensive areas (Perpiña Castillo

et al., 2024). This shift in land use has not only promoted changes in land cover, but also modified the likelihood of human-





caused ignitions in forested areas nearby to these highly populated zones (D'Este et al., 2020; Sjöström and Granström, 2023). On the other hand, unprecedented fire weather and climate factors modulate fire-prone conditions at the spatial and temporal level. The changing spatial patterns of fire regimes are largely a response to the newly emerged extreme climatic conditions and their influence on the arrangement and condition of fuels, affecting the potential for new ignitions and causing spread

patterns that can hardly be confronted by the established fire suppression techniques (Galizia et al., 2021, 2023; Kelly et al., 2023; Pais et al., 2023). Well-known fire danger rating indices like the NFDRS (Schlobohm and Brain, 2002) or the FWI (Stocks et al., 1989) have become essential for daily planning and resource allocation (Resco de Dios et al., 2022; EFFIS, 2018). These indices are often used as a proxy for understanding the expected fuel moisture conditions and fire spread potential (Boer et al., 2017; Resco De Dios et al., 2022; Rodrigues et al., 2023). In turn, seasonal patterns and climate anomalies influence the potential for extreme fire occurrence (Coogan et al., 2020; Rodrigues et al., 2018, 2020).

influence the potential for extreme fire occurrence (Coogan et al., 2020; Rodrigues et al., 2018, 2020).

There are several studies at global (Chuvieco et al., 2021), European (Ochoa et al., 2024; Pettinari and Chuvieco, 2020), regional (Jiménez-Ruano et al., 2022; Rodrigues et al., 2018; Trucchia et al., 2023) or local scales (Vilar del Hoyo et al., 2011) dealing with various aspects of fire danger, such as ignition probability and fire spread. However, most of the analyses carried out to date focus on specific and homogenous regions, except for a few cross-regional studies. Likewise, most of them feature

contrasting scales, methods, and data that hamper the integration of findings into meaningful conclusions (Costafreda-Aumedes et al., 2017). Hence, there is a need to develop cohesive assessments using approaches and techniques that offer comparable outcomes, covering the diversity of bioregions, climates, and fire regimes, to achieve an integrative strategy for fire risk assessment for broader geographic regions, i.e., pan-European (Oliveira et al., 2014).

We hypothesise that, despite the common body of drivers of human-caused ignitions (Chicas and Østergaard Nielsen, 2022;

Costafreda-Aumedes et al., 2017), there are significant differences of their influence across the different European landscapes. To elucidate this assumption, we present an assessment of human-caused ignition probability across five representative European regions (pilot sites, PS) that encompass different wildfire regimes, contrasting environmental and climate settings: PS1- Kalmar Iän (South-East Sweden), PS2- Southern Brandenburg and Eastern Saxony (Germany), North Bohemia (Czechia), and Lower Silesia (Poland), PS3– Central region of Portugal, PS4 – Barcelona province (Spain) and PS5- Attica

region (Greece). We aim (i) to create at the PS scale human-caused fire probabilities spatial assessments to identify areas where new fires are most likely to ignite in the future. These spatial datasets are critical for informing the wildfire spread simulators since they require an ignition probability grid to allocate ignitions for stochastic simulations over broad landscapes and understanding wildfire exposure that can be caused from realistic and potential new fires (Alcasena et al., 2021). In turn, we seek (ii) to provide further insight into the role the drivers of wildfires are playing on ignitions, focusing on unravelling their

relative influence across PS and support risk management. Finally, (iii) we explored the differences between full and local models in terms of performance and influence of the driving forces of ignition, providing a baseline for determining scale effects in ignition modelling and mapping. For this purpose, we developed and validated Random Forest spatial models of ignition probability (for each PS individually and pooling all PS together) based on human, climate and territorial drivers.





## 2. Materials and methods

**2.1 Research context and description of the pilot sites**

The study is framed within the European H2020 project FirEUrisk (Chuvieco et al., 2023). The project contemplates different temporal and spatial scales, the latter being tackled from a three-fold perspective upscaling from small demonstration sites (1:5,000), through pilot sites (1:200,000; the scale showcased here), up to the entire European territory.

The five pilot sites (PS) that we analysed were selected following different criteria representing various climatic and socio-

economic features across Europe (**¡Error! No se encuentra el origen de la referencia.**). In the case of PS-1 in South-Eastern Sweden, we investigate a region with continental climate (Dfb, according to the Köppen climatic classification) with moderate fire activity that is likely to increase under the influence of climate change. PS-2 (also Dfb climate) encompasses four smaller regions across three countries of Central Europe (Germany, Czech Republic, and Poland); we focused on capturing the potential of ignitions that are highly probable to produce a transboundary event, i.e. potential incoming fires from neighbouring

countries that may require a coordinated transnational intervention and management for prevention. PS-3 and PS-4, located in the Central region of Portugal and the Barcelona province (Eastern Spain), depict Mediterranean-type climate conditions (Csb, Csa, Cfa and Cfb). Both pilot sites focus on the influence of Wildland Urban Interfaces (WUI), depicting regions with high fire frequency linked to human pressure on wildlands. Finally, PS-5 (Attica Region, Greece) was selected to understand how the Mediterranean subtropical dry climate (Csa) that its prevalent over a densely populated region (the greater metropolitan

area of Athens), with highly flammable vegetation and dominant annual high-speed winds, produce catastrophic peri-urban wildfires that historically cause a high number of fatalities from large-scale fire events.



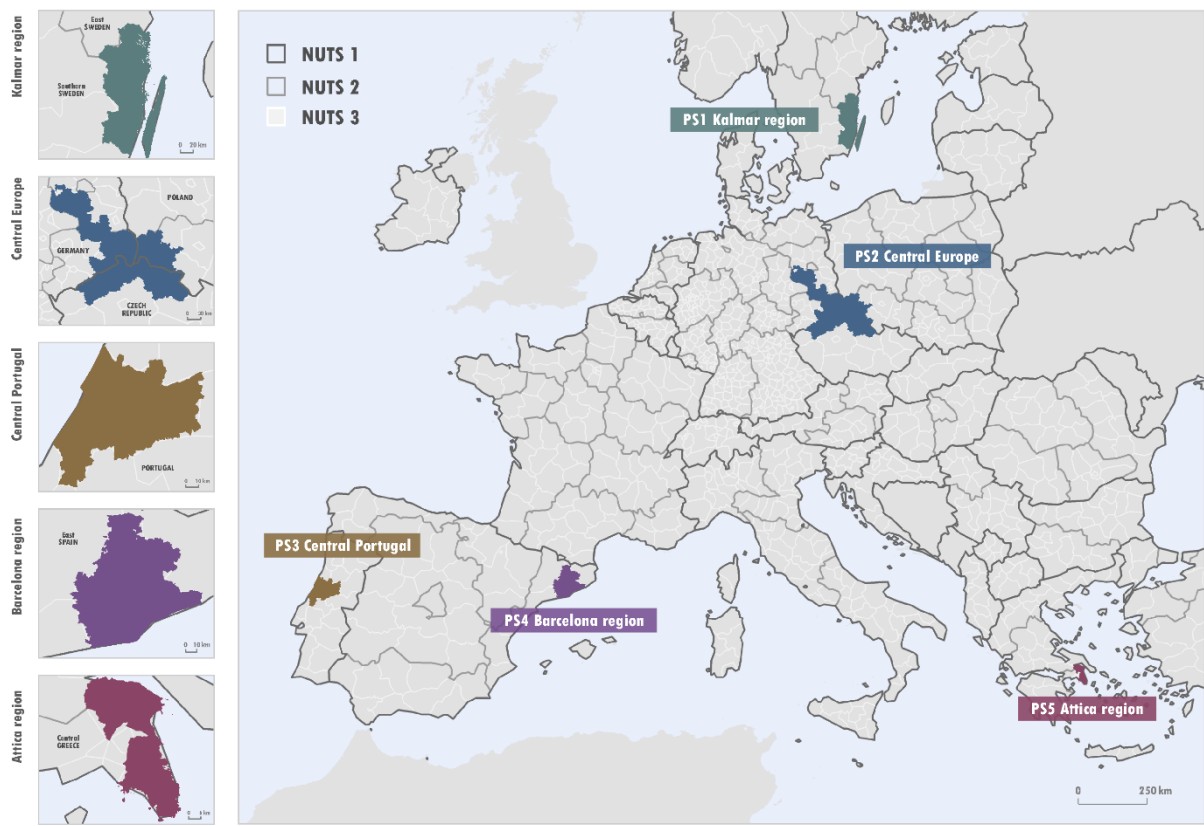

**Figure 1: Locations of Pilot Sites (PS) across Europe with varying fire-regimes and climatic conditions where we modelled human-caused wildfire occurrence.**

## 2.2 Wildfire data and response variable

Models were built from historical fire records of human-related ignitions in the selected PS. We employed eight datasets provided by national and local fire management agencies (Table 1; Figure S1). The temporal coverage varied across regions and/or countries, ranging from 1996 to 2022. We processed and retrieved the coordinates of all fire records attributed with a non-natural cause of ignition, selecting fire events >1 ha to prevent excessive bias and unbalance in sample sizes among regions that resulted from different criteria in fire size reporting or retrieval of the coordinates of the ignition point (San-Miguel-Ayanz et al., 2012).

From these data we created a binary response variable that consists of fire presence (coded as 1) and fire pseudo-absence (coded as 0) locations. Ignition points were used as presence locations, while pseudo-absence locations were spatialised as random points across each PS. The placement region was constrained to burnable land cover types according to the fuel type map (Aragoneses et al., 2023), and outside a buffered area of 500 meters distance from fire ignitions records and other pseudo-absence locations (i.e., absence points 0 were placed within 500 meters distance of a historical ignition or another absence location). The number of pseudo-absence locations was set equal to the amount of fire ignitions in each PS. To evaluate the





potential influence of the pseudo-absence sampling procedure in modelling outcomes, we built 1000 realizations, each conducing to a model realization.


**Table 1. Summary of ignition data per pilot site, country and/or regions (in parenthesis), temporal coverage (first-last year) and total number human-caused fires records.**

| Pilot Site | Country (Region) | Temporal coverage | Number of ignitions | Ignition density (Ignitions/year/km²) |
|---|---|---|---|---|
| PS1 - Northern Europe | Southeast Sweden | 1996 - 2020 | 619 | 0.0022 |
| PS2 - Central Europe | Czech Republic (Karlovy Vary, Ústí nad Labem and Liberec) | 2016 - 2020 | 1636 | 0.0038 |
| | Poland (Silesia) | 2007 - 2017 | | |
| | Germany (Saxony) | 2008 - 2021 | | |
| | Germany (Brandenburg) | 2010 - 2020 | | |
| PS3 - Central Portugal | Portugal | 2001 - 2020 | 1530 | 0.012 |
| PS4 - Barcelona | Spain | 2008 - 2018 | 2133 | 0.028 |
| PS5 - Attica region | Greece | 2017 - 2021 | 188 | 0.031 |

## 2.3 Explanatory variables

We collected a set of variables related to human pressure on wildlands, presence of agricultural activities, accessibility, land cover types and transitions, fuel types and fire-weather. Explanatory variables for the modelling of ignition probability were selected based on literature review (Chicas and Østergaard Nielsen, 2022; Costafreda-Aumedes et al., 2017) and experience acquired from working with regional and national scale human ignition models (Chuvieco et al., 2014; Jiménez-Ruano et al., 2022, 2023; Martín et al., 2018; Ochoa et al., 2024; Rodrigues et al., 2014, 2016, 2018; Rodrigues and De la Riva, 2014).

Variables were created from data sources available for all regions, hence enabling comparison among PS. We reprojected all data to the same coordinate system and converted all explanatory variables into a set of 100 m resolution raster layers to extract geospatial information at fire ignition and pseudo-absence points.





### 2.3.1 Human pressure on wildlands and accessibility

In recent times, there has been an increasing trend in human presence and pressure on wildlands, which can be attributed to
several factors. To analyse the human presence and pressure on wildlands, we used population density, distance to the WUI and distance to roads. The population density data was retrieved from the Global Human Settlement GHS-POP R2023A (Schiavina et al., 2023), a product developed by the European Commission Joint Research Centre, and it is expected to have a positive relationship with human-caused ignitions (Rodrigues et al., 2022). On the other hand, distance to the WUI, which is the Euclidean distance to the boundary layer between wildlands and urban settlements, is derived from the categorical
aggregated Corine Land Cover maps of 2018 (Table S2). Following the literature, we anticipated a higher likelihood of human impact in built-up areas that are closer to the WUI (Costafreda-Aumedes et al., 2017). As a proxy for accessibility, we calculated the distance to roads, defined as the Euclidean distance from all types of roads retrieved from the Global Roads Inventory Project (GRIP) by the GLOBIO model; a higher probability of human impact is presumed closer to roads (Leone et al., 2003).

### 2.3.2 Agricultural related interfaces

Another key factor influencing human-caused fire ignitions is the proximity to interfaces between wildlands and areas of traditional economic activities, specifically agriculture and grasslands/pastures. The distance to the Wildland-Agricultural Interface (WAI) and the Wildland-Grassland Interface (WGI) is crucial in understanding these dynamics (Rodrigues et al., 2014). To delineate these interfaces, we replicated the methodology used for defining the WUI but instead targeted on
croplands and grasslands (Table S1). These interfaces are areas where human agricultural activities meet wildlands, creating zones with an elevated ignition risk due to agricultural practices such as burning crop residues, machinery use, and other land management activities. We expected the risk patterns to replicate those observed for the WUI, where closer proximity to the interface correlates with increased ignition likelihood.

### 2.3.3 Land cover types and land cover transitions

Land cover, particularly the mix of urban, agricultural, and wildland areas, is also a significant indicator of potential fire ignition sources (Costafreda-Aumedes et al., 2017). We calculated the percentage of land covered by urban, agricultural, and wildland classes. Mixed-cover areas, where urban development encroach on wildlands or agricultural lands transition into natural vegetation, are particularly vulnerable. These zones often experience higher human activity levels and potential conflicts, which can lead to accidental or intentional ignitions. Furthermore, we calculated two land cover transitions:
urbanization, i.e., any land cover change from natural vegetation into urban areas; and forest expansion, any transition into areas with natural vegetation (Probeck et al., 2021).



### 2.3.4 Fuel types

The availability and type of fuels are fundamental in determining wildfire behaviour and ignition likelihood. We adapted the procedure proposed by Aragoneses et al. (2023) to create a spatial approximation of fuel types (Table S2) by integrating several high-resolution datasets. These included the Corine Land Cover 2018 (CLC; European Environment Agency, 2019) and CLC+ Backbone resampled to 100 m (Probeck et al., 2021), Tree Cover Density (TCD; European Environment Agency, 2020) at 100 m from the European Environment Agency, Global Forest Canopy Height resampled to 100 m (Potapov et al., 2021), and the burned areas from 2018 to 2022 retrieved from the European Forest Fire Information System (EFFIS). By reclassifying and merging these datasets, we developed a detailed map of fuel types across each study area, as shown in Table S1, which outlines the correspondence between FirEUrisk defined fuel types and the reclassification criteria used.

### 2.3.5 Fire weather

Weather conditions influence the probability of ignition mainly by modulating the moisture content of the dead fraction of fine fuels (Rodrigues et al., 2023; Van Wagner, 1987). We calculated the dead fine-fuel moisture content (DFMC) following the method by (Rodrigues et al., 2024). The method is based on empirical relationships between Vapor Pressure Deficit (VPD) and DFMC. We calculated daily DFMC from temperature and relative humidity acquired from reanalysis data from the ERA5 Land product (Copernicus Climate Service, 2017). We aggregated daily DFMC into annual products using data inside the main wildfire season (June to September). We calculated the 5th percentile of DFMC and the DFMC anomaly (expressed as Z-Scores), in the period 1991-2021.

### 2.4 Modelling approach

Random Forest (RF, Breiman, 2001) binary classification models were calibrated from historical fire records and ignition drivers. The effectiveness of machine learning algorithms is widely recognized in fire modelling (Kim et al., 2019; Milanović et al., 2021; Oliveira et al., 2012; Sebastián-López et al., 2008). Among the wide range of algorithms available, RF stands out as a trade-off between efficiency, simplicity in its calibration and optimization, and versatility in its application (Bar Massada et al., 2012; Chicas and Østergaard Nielsen, 2022; Rodrigues and De la Riva, 2014). RF is a tree-based ensemble regression and classification algorithm that uses a bagging strategy to create and merge a set of decision trees. The modelling approach to assess ignition probability is three-fold: first, we calibrated individual models for each PS. Then, we calibrated the so-called full model, pooling all PS data together to mimic a region-wide model. Finally, we compared individual models with the full model to delve into potential differences in model performance and the influence of ignition drivers.

### 2.4.1 Model calibration

We fitted 1000 RF model realisations (one per pseudo-absence sample) per PS, using the RF implementation in the *ranger* R package (Wright and Ziegler, 2017). We implemented a bootstrapping procedure (1000 model realizations) retaining 80% of



observations to train a model and the remaining 20% for testing their performance. RF hyperparameters were optimised; to do so we tested a set of values for each hyperparameter over a null model by means of repeated cross validation with 10 partitions and five repetitions. Concretely, for *mtry* (number of predictors used at each branch split of a decision tree) we tested values from three predictors to number of predictors -1; for *min node size* (a stop rule based on the minimum number of observations in a terminal node that controls tree depth) values ranging from 5 to 90% of observations were tested; and the *splitrule* (the criteria to select a value of *mtry*) was the Gini index used as a rule of thumb in classification assessments. The criteria to define the best hyper-parametrisation was the highest mean Area Under the Receiver Operating Characteristic Curve (AUC) across the different repeats (see section 0). To account for spatial autocorrelation, we added an Autocorrelation Control (AC) term calculated as the distance to the center and to the four corners of the study region, and also x,y coordinates of each point. Additionally, we calibrated the full model. This model incorporates a dummy variable, i.e., the PS code, to control potential effects from the geographical origin of the observations. On top of AC term, we implemented a semivariogram to estimate the minimum distance between observations during the sampling procedure. The full model provides a baseline for further comparison with PS models, acting as proxy of a regional model. Specifically, we compared the rank in importance of model covariates with PS models and evaluated the differences in the accuracy of the predictions between the full model and each PS model by comparing performance metrics.

### 2.4.2 Model validation and performance of the explanatory variables

The predictive accuracy of the models was assessed by calculating the AUC using the test samples from each model realization (Turner, 2020). The AUC is a threshold-independent metric based on plotting the true positive rate against the false positive rate along the continuum of probability values (0-1). An AUC above 0.70 is deemed as moderately good, while 0.80 marks the threshold of sufficiently performance (Metz, 1978). The influence of the predictors in the probability of ignition was measured in terms of variable importance (increase in node impurity), expressing it as a relative value between 0 to 100%. Dependence plots relate the predicted response (probability of ignition) with the range of values of a predictor. The shape of the profiles informs about the type of relationship (linear/non-linear, positive/negative). Finally, to ensure model reliability, we analysed model residuals' spatial autocorrelation using the Moran's I index (Moran, 1950).

We used the R language for statistical computing (R Core Team, 2024) and its most widely used Integrated Developed Environment RStudio, to carry out the entire process. Concretely, we used functions from different packages such as *tidyverse* (Wickham et al., 2019) for data management and visualisation, *caret* (Kuhn, 2008) to optimize model hyperparameters, *ranger* (Wright and Ziegler, 2017) as a fast implementation of RF model, *sf* and *terra* (Hijmans, 2023; Pebesma, 2018) to manage spatial data (vectorial and raster), *pROC* (Robin et al., 2011) for model evaluation and *ape* (Paradis and Schliep, 2019) for spatial autocorrelation analysis.





## 3. Results

### 3.1 Predictive performance and effects of spatial autocorrelation

The capacity of the individual models to predict ignitions varied across pilot sites (Figure 2; Figure S2**¡Error! No se encuentra**

**el origen de la referencia.**), with AUC values ranging from 0.70 (East Attica) to 0.89 (Central Europe). The performance of the full model stood close to the average of the PS's models (AUC=0.81). However, its capacity decreased when evaluated at PS level, ranging from 0.61 to 0.85. Again, East Attica attained the lowest accuracy and Central Europe the highest, but it is remarkable that the only PS with AUC above the 0.80 threshold was the latter, with Central Portugal and Barcelona stepping down below 0.80 and East Attica and Southern Sweeden below 0.70.

Most models showed no spatial structure in the residuals (Moran's I non-significant p>0.05; Figure 3), meaning that they can be further used to draw inference based on model outcomes. The AC control successfully alleviated spatial autocorrelation, ranking remarkably high in Central Europe and Central Portugal (among top 5 predictors, Table 2). Disregarding the AC control in the full model led to spatially correlated residuals in all models.





**Figure 2. Frequency distribution of performance in model predictions (n=1000). Each model corresponds to a random sample of the ignition absence locations.**





**Figure 3. Frequency distribution (n=1000) of Moran's I index calculated from the residuals in each model realization. The dashed vertical line indicates the p<0.05 threshold, below which residuals can be considered as correlated with 95% confidence.**


## 3.2 Main explanatory factors of ignition and regional differences

The spatial distribution of human-caused ignition probability was modulated by the combination of different variables in each PS. Table 2The influence of the predictors in terms of relative importance (i.e., the relevance of a given variable in reaching a prediction) and type of relationship (i.e., whether it boosts or hinders the likelihood of ignition; Figure S3 and Figure S4). Fires





in southern Sweden (PS1) tend to start under dry and warm seasons (DFMC$_{zs}$), preferably in forest-dominated areas or natural landscapes near human settlements such as the WUI, with a greater emphasis on forestry and less on agriculture or open grasslands (away from WAI and WGI). These regions combine proximity to human activity (roads, WUI) and are situated in forested areas where agricultural activities are minimal, making them more prone to wildfires due to temperature anomalies and human activity rather than agricultural ignition sources. In Central Europe (PS2), fires started under abnormally low DFMC

with a significant fraction cover of wildlands, such as forests or natural landscapes, that provide abundant fuel. The chance of ignition increases in zones close to the WUI, where human settlements meet natural areas. Additionally, proximity to roads further elevates the probability of ignition. In the Central Portugal (PS3), densely populated areas with abnormally low DFMC boosted ignition probability. Higher probability was attributed to regions located at a moderate distance away from WUI, where rural communities and forests intermingle. Lastly, proximity to roads further increased ignition probability. In the

Barcelona region (PS4), wildfire ignition was boosted by proximity to roads and the WUI. These areas become especially vulnerable with increased population density, which raises the chances of fire ignition from human activities. A DFMC anomaly further raised the fire danger. Additionally, proximity to the WAI compounds the risk of ignition. In the Attica region (PS5), areas particularly susceptible to wildfires were characterized by proximity to the WUI, where urban and suburban areas stand near moderate wildland presence, such as forests or scrublands (% wildlands). Again, the proximity to WAI fosters the

potential for ignitions from farming activities or equipment. Lastly, population density in these areas can exacerbate the risk of human-caused ignitions.

When modeling all PS together, regions at heightened probability of ignition were characterized by DFMC anomalies accompanied by their yearly average. Locations near roads and the WUI were especially prone to ignition. Furthermore, population density was a critical factor, followed by the proximity to agricultural activities (WAI). The full model seemed to

provide an averaged version of PS models with reinforced importance of fire weather factors, being the only model that promoted mean DFMC among the top tier drivers. Moreover, the full model disregarded certain drivers like the fraction covered by wildlands, which played a contrasting role in some PS. On the other hand, neither land cover-related features such as the fraction of urban and agricultural lands or dominant fuel types, nor the land cover transitions like urbanization and forest expansion, played a determinant role or contributed significantly in any model.


**Table 2. Relative importance of the predictors. Numbers indicate the rank in importance. Symbols between parenthesis relate to the explanatory sense (when applicable): (+) positive relationship, (-) negative relationship, and (o) flat profile/non-meaningful. The top 5 variables in importance are highlighted in grey.**

| Factor | Variable | All | PS1 | PS2 | PS3 | PS4 | PS5 |
|---|---|---|---|---|---|---|---|
| Fire weather | DFMC$_{zs}$ | 1 (-) | 1 (-) | 1 (-) | 2 (-) | 4 (-) | 9 (o) |
| | DFMC$_{avg}$ | 2 (-) | 7 (-) | 6 (-) | 9 (-) | 9 (o) | 10 (o) |
| Wildland interfaces | Dist. WUI | 5 (-) | 2 (-) | 4 (-) | 3 (+) | 1 (-) | 1 (-) |
| | Dist. WAI | 6 (-) | 4 (+) | 7 (-) | 7 (o) | 5 (-) | 4 (-) |





|  |  |  |  |  |  |  |  |
|---|---|---|---|---|---|---|---|
|  | Dist. WGI | 7 (o) | 5 (+) | 8 (o) | 6 (o) | 6 (+) | 3 (+) |
| Human pressure | Dist. roads | 3 (-) | 3 (-) | 5 (-) | 4 (-) | 2 (-) | 6 (-) |
| and accessibility | Pop. Density | 4 (+) | 11 (o) | 11 (o) | 1 (+) | 3 (+) | 5 (+) |
|  | %Wildlands | 9 (+) | 8 (+) | 2 (+) | 8 (-) | 8 (o) | 2 (-) |
| Land cover | %Agriculture | 10 (o) | 9 (o) | 9 (-) | 10 (+) | 11 (o) | 7 (+) |
|  | %Urban | 12 (o) | 12 (+) | 12 (+) | 12 (o) | 10 (-) | 11 (+) |
| Fuel types | Fuel type | 11 | 10 | 10 | 11 | 12 | 12 |
| LC transition | Urbanization | 14 (-) | 13 (o) | 14 (-) | 13 (o) | 13 (o) | 13 (+) |
|  | Forest exp. | 15 (+) | 14 (o) | 13 (o) | 14 (o) | 14 (+) | 14 (o) |
| Dummy | PS identifier | 13 | - | - | - | - | - |
| variables | Autoc. Contr. | 8 | 6 | 3 | 5 | 7 | 8 |

## 3.3 Spatial patterns of ignition probability

The unique combination of driving factors at the PS level produced a singular spatial pattern of ignition probability in each of them (Figure 4). The backbone of each pattern lay in human factors, with DFMC set as the average conditions to ensure the production of comparable maps. For example, in south-eastern Sweden we can observe two main clusters of probability in the mid sector, with a decreasing gradient towards the south. In Central Europe, the German part of the region displayed higher probability, slightly spreading across the center in the border between countries. In central Portugal, the high ignition probabilities were clustered in densely populated zones, like Barcelona, which higher likelihood in the WUI areas around the metropolitan region of the capital city. In Attica (PS5), the driving forces behind the spatial distribution of probability showed the strongest link with human activities, with probability mirroring the distribution of urban and agricultural lands coalescing with wildlands.

When applying the full model to map the probability of ignition, the general pattern appears to prevail, although differences were observed. The distribution becomes more diffused, with higher probabilities attributed to a larger number of pixels. Overall, the predicted probability was higher when the full model was used. However, some small enclaves were underestimated. Therefore, the fine-grained pattern becomes less accurate, as we already established when validating the full model at the PS level. Looking specifically at the different Pilot Sites, PS1 and PS2 showed minimal differences between PS model and full model, as both were highly influenced by DFMC. In contrast, PS3 and PS4, located in Mediterranean regions, highlighted the importance of human factors like population density (PS3) and distance to WUI (PS4), which made local models more overpredict compared to the full model. PS5, with limited data, showed a tendency for the local model to predict higher ignition probabilities, reflecting the dominance of site-specific patterns.







**Figure 4. Spatial distribution of ignition probabilities predicted with PS models (left column) and the full model (middle columns), and difference in the prediction (right column).**

## 4. Discussion

This ignition probability assessment of anthropogenic wildfires was built with a set of predictive models in five distinct study sites across Europe. The ultimate drivers of human-caused fires are known to vary by country and even locally by region

(Oliveira et al., 2014), resulting in diverse modelling schemes and strategies that hinder our ability to draw joint conclusions (Costafreda-Aumedes et al., 2017). The differences in comparable models may offer enhanced information about each pilot site, thus better guiding management strategies across larger regions.

### 4.1 Driving forces across regions

Our results suggest that fire weather conditions (Keeping et al., 2024; Resco De Dios et al., 2022), accessibility (Costafreda-

Aumedes et al., 2017; Guo et al., 2016; Vacchiano et al., 2018) and human pressure on wildlands (San-Miguel-Ayanz et al., 2012) were the leading influencing factors in most models (Table 2). The full model revealed a systematic pattern of human-caused fires linked to abnormally low DFMC, starting in densely populated areas close to accessible urban settlements, or in intensive agricultural lands. However, the importance of these drivers varied across sites, and nuances between the analysed regions were manifold.

In the case of Southern Sweden (PS1) and Central Europe (PS2), fires occurred preferably during abnormally dry years. Several studies conducted in Sweeden revealed that a remarkable percentage of the burned area (40%) originated from ignitions sparked by the forestry machinery (Sjöström et al., 2019). Road density was also positively related to human-caused ignitions for all Sweden (Pinto et al., 2020), even though these authors found that the population density was the most relevant factor in contrast to our findings, which favours the proximity to the WUI. In Central Europe, other studies have pointed to the

confluence of wildland urban interface, and road density (Ciesielski et al., 2022; Kolanek et al., 2021; Mozny et al., 2021). In fact, in this area of Europe, concretely in Silesia (Poland), the majority of human caused wildfires is related to the burning of stubble by farmers (Ganteaume et al., 2013).

Barcelona and Central Portugal (PS3 and PS4) feature a similar pattern dominated by the joint influence of accessibility and presence of people, a behaviour also evidenced by other authors (Martín et al., 2018; Parente et al., 2018). . However, ignitions

in Portugal seem to be more associated with rural enclaves located at moderate distances from the WUI, while fires in Barcelona and the other pilot sites tend to start in the vicinity of the WUI and road networks, such as along the north-to-south axis of the C-16 highway. These PS were less influenced by DFMC variations, especially Barcelona, given their geographical distribution within Mediterranean-type conditions.

Finally, in the Attica region (PS5), fires were strongly tied to a mix of human-related factors, with a modest climate

contribution, probably due to mostly favourable conditions due to its Mediterranean climate. Results pointed at the proximity to the WUI as the main factor followed by accessibility by road, especially in combination with degraded or recently abandoned





wildlands linked to livestock decline (Colantoni et al., 2020). Fires were also linked to agricultural activities, whereas densely populated enclaves boosted ignition potential as well, including the two major fires in the past three years, each exceeding 8,000 hectares, sparked by powerline.

Implementing effective fire ignition prevention policies is crucial for reducing human-caused wildfires. Particularly impactful strategies include public education to promote responsible behaviours, fire use restrictions during high-risk periods, strict enforcement of regulations to deter negligence, and vegetation management along high-traffic corridors and in WUI areas to reduce hazardous fuels. Anticipating fire-prone weather conditions is key to identifying temporal windows during which to restrict the use of fire or reinforce public risk awareness in all regions. In northern and central Europe, focusing on managing

vegetation near WUI zones and roads is essential, and proactive measures such as early warning systems and public advisories during high-risk periods can significantly reduce accidental ignitions. In the western Mediterranean, reinforcing territorial planning to regulate urban and rural expansion into forestlands is crucial. Implementing land-use policies that prevent uncontrolled development in fire-prone areas can mitigate ignition risks and enhance landscape resilience. In highly fire-prone regions like Attica, promoting responsible recreational use of forests is necessary to prevent fire occurrences. Likewise,

advocating responsible land-use practices and enhancing community engagement in fire prevention efforts can address the human activities contributing to ignition risks.

## 4.2 Profiles and type of relationships

All drivers displayed distinct non-linear profiles in their relationship with the probability of human-caused fire ignition. The variables that demonstrate the greatest variability in probability values were the distances to roads, the WUI, and the WAI.

There is a notable sharp decrease in probability values, leading to higher probabilities observed near roads, urban discontinuous areas, and crop patches near wildland. This effect occurs because roads offer accessibility and serve as sources of ignition, alongside agricultural and discontinuous urban areas (Ganteaume et al., 2013).

It is important to keep in mind that socioeconomic and demographic variables depend on external factors such as cultural differences and can have complex relationships which can be masked in global models. For instance, the negative relationship

between fire ignitions and distance to roads identified in previous studies likely reflects human accessibility to easily ignitable fuels and vegetation close to roads (Chicas and Østergaard Nielsen, 2022; Costafreda-Aumedes et al., 2017; Guo et al., 2016; Vacchiano et al., 2018). Many roads pass through degraded landscapes where natural vegetation, especially trees, has been removed for road construction and maintenance. This removal leaves flammable grasses and small Mediterranean shrubs that can easily ignite regardless of their potential for propagation into a large fire event. Since most fires in our research were small

(<100 hectares), the positive impact of roads on ignition outweighs its negative effect on fire spread, which only becomes significant when focusing on large fires as presented by Ochoa et al., (2024). These differences stress the importance of tailored preventive fuel management around key hot spots along the road network.

The percentage of wildland plays a contrasting role depending on the environment. In southern areas, it significantly contributes to human-caused ignition probability when wildland proportions are lower. Conversely, in northern areas, higher



proportions of wildland lead to an increase in ignition probability. This is correlated with fuel moisture and the heightened ignition capacity of drier environments (Jurdao et al., 2012). Several studies highlight population distribution as a critical variable in modelling human-caused ignitions (Ciesielski et al., 2022; Costafreda-Aumedes et al., 2017; Ganteaume et al., 2013; Martín et al., 2018), exhibiting a sharp response curve that flattens abruptly beyond a close breakpoint. This suggests that even a minor presence of people may contribute to higher ignition likelihood.

Finally, despite other research, such as the findings by Rodrigues et al., (2019a), establishing a strong relationship between fuel type and ignition probability, the present study found that fuel type played a limited role in the probability of human-caused ignitions. This low importance can be partly constrained by the influence of other land cover related factors such as percentage of wildland, agricultural or urban areas, and the prominent role of DFMC. In other words, fuel types might be redundant when land cover types or fuel moisture content are already accounted for.

### 4.3 Implications for fire danger modelling

Beyond the differences in driving forces, several lessons can be learnt from our modelling procedure: modelling at regional scales limit the capacity to capture local patterns, failing to capture the fine-grained patterns of probability.

The full model was successful in terms of performance, standing at an intermediate level compared to its PS counterparts. However, when evaluated locally, performance dropped in all PS, rendering its predictions unreliable in south-east Sweeden

and the Attica region by decreasing AUC by 0.10 down below 0.70 (Metz, 1978). This drop in performance is likely due to differences in the contribution of predictor variables among models, hindering specially those situations more difficult to generalise because of data scarcity (Harrell, 2015). As a side-effect, the patterns of probability were different, with a tendency to overestimate the chances of ignition and driving sharp changes in the local patterns.

The effects of spatial autocorrelation in models are often overlooked. Recent reviews in ignition model and wildfire

susceptibility summarizing the efficacy of algorithms and/or the driving factors behind wildfire incidence disregarded in their evaluations whether modelling attempts explicitly accounted for spatial autocorrelation (Chicas and Østergaard Nielsen, 2022; Costafreda-Aumedes et al., 2017). Spatial autocorrelation refers to the phenomenon where the values of a variable exhibit a correlation based on their geographic proximity, meaning that sites located near each other tend to have similar (or related) values for that variable (Tobler, 1970). There is evidence that neglecting spatial autocorrelation affects the precision of the

estimates (Guélat and Kéry, 2018)(Table S3). The spatially clustered structured of wildfires is long known (Chou, 1992). Consequently, accounting for its potential influence is key to deem if the spatial pattern in the dependent variable could be explained by the spatial pattern observed in the predictors (Dormann et al., 2007; Oliveira et al., 2012). We integrated spatial autocorrelation including an AC term in the models (Behrens et al., 2018), which successfully controlled the spatial structure in model residuals in 65% of the full model realizations and in 80% of the PS models. The AC term ranked high in Barcelona

and Central Portugal, indicating a particularly clustered pattern of ignition.



### 4.4 Limitations and further improvements

Despite the valuable outcomes we achieved, the study presents limitations that should be mentioned. Firstly, the strong influence of sample size on model performance is particularly crucial in the case of PS5, where results exhibited the lowest accuracy. The present approach is based on static human factors, but further research would be necessary to incorporate daily-scale (e.g., commuting) and seasonal (e.g., tourism and recreational activities) dynamics of human behaviour into more sophisticated models. However, we have reached a compromise between the accuracy of the models and the minimal and less complex number of fire factors needed for modelling. Finally, the present analysis is a first step towards more extrapolated prediction procedures to obtain general and comparable human ignition models for the rest of the European territory, or even other similar regions of the globe with similar intrinsic human factors.

### 5. Conclusions

In this study we developed predictive models to assess the probability of human-caused fires across five European regions, each characterized by its own ignition factors and fire regimes. The results highlight the central role of weather conditions, accessibility and human pressure on wildlands as key factors, although the importance of these factors varies among regions. Proximity to roads, WUI and population pressures were consistently influential. Fire-weather anomalies were especially important in northern regions, while human factors gain importance in the Mediterranean. Random Forest has proven to be a powerful tool for predicting wildfire ignition probability, achieving high accuracy with AUCs above 0.80. In addition, spatial autocorrelation, often overlooked in wildfire models, was successfully integrated, controlling spatial patterns and improving model reliability. These results underscore the need for regionally tailored ignition prevention strategies and the importance of accounting for human-caused ignition patterns in landscape-scale stochastic fire modelling.

### Acknowledgments

This work was financed by the projects FirEUrisk – Developing a Holistic, Risk-wise Strategy for European Wildfire Management, which received funding from the European Union's Horizon 2020 research and innovation programme under grant agreement No. 101003890; FIREPATHS (PID2020-116556RA-I00), funded by the Spanish Ministry of Science and Innovation (MCIN/AEI /10.13039/501100011033); and FireCycle (CNS2023-144228), funded by the Spanish Ministry of Science and Innovation (MCIN/AEI/10.13039/ 501100011033). The authors express their gratitude for receiving the contract 'Margarita Salas' (MS-240621), held by Adrián Jiménez-Ruano and granted by the Spanish Ministry of Universities.



**Data availability**

All raw data will be available in Zenodo repository in the final version of the manuscript.

**Author contributions**

JC, FD, and HM planned the campaign; JC, FD, HM, AF, and DW performed the measurements; JC, FD, AF, and DW analyzed
the data; JC and FD wrote the manuscript draft; AF, MEGH, HDvdG, and TR reviewed and edited the manuscript.

**Competing interests**

The authors declare that they have no conflict of interest.

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
