# Peer review of "Assessing human-caused wildfire ignition likelihood across Europe"

_EGUsphere, 2025_

## Author Comment (AC1)

**#RC1**

**Overall**

The paper presents a spatially explicit modeling approach for human-caused wildfire ignitions across various European regions. The study applies machine learning techniques, specifically Random Forest models, to analyze historical fire records and environmental variables such as land cover, population density, accessibility, and dead fine-fuel moisture content (DFMC). The results highlight that the most influential variables in predicting ignition probability are DFMC anomalies, proximity to the Wildland-Urban Interface (WUI), and road accessibility.

The study emphasizes the role of anthropogenic factors in fire ignition and provides valuable insights into human-caused wildfire ignitions. However, it faces challenges related to model generalizability, temporal dynamics, and policy application.

Thank you for your valuable comments. We have carefully addressed all your observations, and all replies are highlighted in **red** color. Additionally, we have incorporated a Supplementary Material section to provide further details to clarify data providers and overfitting analysis.

**Major comments**

- One major comment regarding this paper is that a big portion of the core of this study has been already published as a conference paper https://doi.org/10.23919/SpliTech58164.2023.10193249 with a high degree of overlap in content, methodology, and key findings of the current article under review. Here it seems that there is an extended and refined version of the previously published material, thus I leave it up to the editors to make a decision about that.

We appreciate the reviewer's observation. The conference paper the reviewer refers to was an early proof-of-concept focused on pilot sites with a preliminary dataset,limited predictors and incomplete formal analyses. The current manuscript substantially extends this work by expanding to a European-scale model, incorporating methodological advances such as the autocorrelation control (AC) term, DFMC mean and anomalies, and new anthropogenic variables, systematically comparing site-specific and full models. It also provides comprehensive cross-validation, performance analysis, and interpretation of spatial ignition patterns, while discussing broader implications, limitations, and operational applications. The conference paper is thus an early-stage presentation of one component, whereas this submission constitutes the complete study. This conference paper was already cited in the manuscript.

- Although the study acknowledges spatial autocorrelation effects,  does not fully resolve them, leading to reduced model performance in regions with fewer fire records (e.g., Attica). This undermines the reliability of the model when applied to areas with limited historical data, reducing its effectiveness for wildfire prediction.

We appreciate this observation. To address spatial autocorrelation (SAC), we incorporated an autocorrelation control term into all models, which reduced Moran's I to non-autocrrelated levels in most realizations. However, in data limited regions like Attica, some residual SAC persisted, likely contributing to lower AUC values. We acknowledged that limited historical ignitions constrain generalization and promote overfitting in lines 420-423

- In this study the authors develop separate models for different pilot sites and then compare them to a full model. While this approach helps capture local variations, it may lead to overfitting within specific regions, limiting the model's ability to generalize ignition likelihood across broader areas. Although the authors discuss some of these aspects (e.g., Section 4) they could provide a more detailed discussion and clearly state all the limitations of their approach.

Thank you for your observation- To assess potential overfitting, we compared the AUC obtained from Out-of-Bag (OOB) predictions—model predictions with reserved sample for OOB error calculation —with the AUC based on independent test samples. As expected, the OOB AUC was consistently lower than the test AUC across pilot sites. This result aligns with previous evidence showing that OOB-based AUC tends to underestimate model performance (https://doi.org/10.1371/journal.pone.0201904). The only exception was PS5, where approximately half of the models showed test AUC < OOB AUC, suggesting some degree of overfitting. We attribute this not to the inclusion of EDF per se, but to the reduced sample size available at PS5, which increases the instability of model estimates (please see figure below). In this sense we caution on these issues in sections 3.1(L253-254) and 4.4.1 (L423-424) and added the below figure in supplementary material.

[Figure]

- For further improvement: Although the authors state some of these issues in Section 4.4, their study focuses on static environmental and anthropogenic variables but does not incorporate seasonal or real-time human activity variations (e.g., increased tourism in summer, agricultural burning periods). Since human behavior significantly influences fire ignition, integrating temporal dynamics would improve model accuracy.

In lines 426 to 430 we acknowledge that seasonal human activities, such as tourism peaks or agricultural burning periods, can influence ignition patterns. However, since exact ignition dates are not known for some pilot sites, we cannot include variables tied to specific days or seasons, just as we could not incorporate daily DFMC. Additionally, our objective was to develop a spatially explicit model, where all predictors vary continuously across the study area. Most seasonal or real-time human activity variables are represented by unique values, lacking spatial variation. Including such non-spatial predictors would compromise the spatial resolution of the model and distort cross-site comparisons. For this reason, we focused on spatially explicit variables available consistently across Europe.

- Although the temporal coverage is short in most areas, did the authors consider any temporal trends in DFMC?

Yes, we did. While the temporal coverage was indeed short in some pilot sites, we accounted for temporal patterns in DFMC by including the anomaly relative to the DFMC baseline as predictors. This allowed us to capture short-term deviations from typical seasonal conditions, which are often critical for ignition occurrence. Given the limited number of years available in certain regions, modelling long-term temporal trends was not feasible without compromising robustness.

**Minor comments**

- Line 28: AUC abbreviation is not introduced earlier.

Done

- Line 105: Needs to be revised-error message.

Corrected

- Line 115: Maybe "seasonal" instead of "annual"?

Changed

- Lines: 104-116: Some references to the related statements are necessary here.

Added

- Lines 128-132: The native resolution of the fuel type is missing here.

The fuel type layer does not have a single native resolution, as it is derived from a combination of several products. The final resolution, consistent with the other variables, is 100 m. We clarified this in the revised version (section 2.3.4 – L183-189).

- Lines 183-185: Could the authors be more specific about the terms reclassifying and merging? Does this also involve any regrid method and if yes, which one?

We aggregated the data using majority vote for categorical variables and mean for numerical variables. In the case of the Global Forest Canopy Height product, used to generate fuel models, which is originally at 30 m resolution, we aggregated it to 90 m using the mean value and then resampled it to match the 100 m grid defined for the other variables. Please see further clarifications in section 2.3.4, L183-189.

- Line 191-194: Is this daily-mean or daily-max DFMC? Could you please clarify what do you mean by aggregating daily values to annual products? Furthermore, could the authors specify the time scale of the 5th percentile and the anomalies? Are these multi-year daily climatological values or something else?

We thank the reviewer for this question. To clarify, we calculated the monthly-mean DFMC values, derived from Monthly reanalysis data from ERA5 (https://developers.google.com/earth-engine/datasets/catalog/ECMWF_ERA5_MONTHLY), which were then aggregated into annual products. Additionally, we computed the mean DFMC over the period 1991–2021 and the annual anomalies (Z-scores) to capture interannual variability. Regarding the 5th percentile, we acknowledge that this was mistakenly mentioned in the manuscript and has now been removed. Please see changes in section 2.3.5 - L198.

- Lines 239-240: Needs to be revised-error message.

Done

- Line 258: needs to be revised.

Done

- Lines 259-276: Could authors provide some further explanation for the limited importance of DFMC in PS4 and especially in PS5? Is this related only to the more frequent low DFMC conditions compared to the northern sites?

We believe so. As the reviewer noted, in the Mediterranean basin low DFMC conditions are very frequent in summer time, which reduces their explanatory power in the models. In addition, other human-related drivers exert a stronger influence on ignition patterns in the Mediterranean region, in contrast to northern Europe where climatic variability plays a more dominant role. Please see new clarifications in section 3.2, L287-290.

- Could the authors provide some explanation for the limited role of fuel type as a predictor? The study finds that fuel type is not a significant factor in human-caused ignitions, which contradicts existing research. This could indicate potential data quality issues or model design limitations. A sensitivity analysis on fuel-related variables would clarify this discrepancy.

Fuel type is not widely used in fire susceptibility research. Concretely, this review paper (Chicas, Østergaard and Nielsen J., 2022. – www.doi.org/10.1007/s11069-022-05495-5) points out that fuel type is used as predictor only in 5 research articles from the 94 analyzed and only in 3 has explanatory power.

In our case, the main causes of ignition in southern Europe are primarily driven by anthropogenic actions and tend to occur near roads and in agricultural-forest interface areas. This results in ignitions predominantly occurring in the same fuel types (grasslands, shrublands, or areas with low tree density, dominated by fine fuels which facilitate ignition).

Regarding modeling, fuel models did not provide significant explanatory power, similarly to findings in another study (Gelabert et al., 2024 - https://www.tandfonline.com/doi/full/10.1080/19475705.2025.2472864). Fuel types are more used to predict fire spread rather than ignition, as ignitions mainly occur in areas with fine fuels.

- The study uses multiple terms for similar human-related ignition factors (e.g., "human pressure on wildlands," "accessibility," "population influence"). Standardizing terminology throughout the paper would improve clarity and coherence.

We appreciate the reviewer's suggestion. However, the terms used in the manuscript are deliberately employed to represent standardized causal factors that have been widely recognized in the literature. Each term reflects a conceptual category that can be characterized through different variables. This approach follows established frameworks, such as those proposed by Leone et al. (2003) (http://dx.doi.org/10.1142/9789812791177_0006) where multiple human-related drivers are grouped under broader thematic concepts.

- Figures illustrating ignition probability (Fig. 4) distributions lack sufficient annotation or explanation (e.g., annotations of subfigures). Enhancing the clarity of these visuals would make the findings more accessible. Furthermore, the colobar for the probabilities could be revised to better communicate the results.

In the revised manuscript, we have incorporated a new version of the ignition probability maps (Fig. 4) in which the results are represented by quintiles. This approach improves interpretability and highlights the relative distribution of probabilities more clearly. We have also revised the figure annotations and subfigure labels to enhance clarity, and updated the colorbar using breaks to better communicate the probability values.

---

## Author Comment (AC2)

**#RC2**

**Reviewer Comments**

The authors in this paper present a comprehensive and methodologically robust assessment of human-caused wildfire ignition probability across diverse European landscapes. By combining machine learning techniques, specifically Random Forest models, with high-resolution geospatial and socio-environmental data, they deliver both localized and regionally integrated ignition probability models. The overall quality of writing is good, with a clear structure, appropriate referencing, and a sound methodological framework.

The topic is highly relevant and timely, particularly given the increasing wildfire risk under changing climate and land use dynamics in Europe. Importantly, the authors' focus on the interplay between local ignition drivers and their generalization into a full model provides valuable insight into the complexity and variability of fire ignition processes. This shift from local to pan-European modelling is of great significance for the development of integrated fire management strategies at the EU scale.

Overall, the manuscript is a solid contribution to the scientific understanding of ignition patterns and offers operationally meaningful outcomes for fire risk management and prevention across Europe.

Thank you for your thorough and constructive comments. We have carefully addressed all your suggestions and concerns. All our responses are highlighted in **red** for clarity. Furthermore, we have added a Supplementary Material section that provides additional details on data, methods, and overfitting analysis.

Below I provide a series of detailed comments and questions that may help the authors strengthen the manuscript even further:

**Specific Comments and Questions**

- **Section 2.3.3**: How did you identify the so-called "mixing areas" algorithmically? Some additional details about the method used would be appreciated.

    We did not explicitly delineate "mixing areas" as unique classes. Instead, for each cell we calculated the percentage of forested, agricultural, and urban land cover. To better capture the effect of mixed-use zones and the associated increase in human activity, we also included in the model the interfaces between wildland and urban, agricultural, and grassland areas. To avoid reader confusion, in the manuscript we substitute the term by *intermix of* by *presence of urban, agricultural...*(L173)

- **Section 2.3.4**: What are the FirEUrisk fuel classes used in the study? Can these be differentiated enough to capture important distinctions in land cover such as eucalyptus in Portugal, which, despite being a broadleaf, behaves quite differently due to its high flammability? Also, how did you project a 10-meter resolution (CLC+ Backbone) raster to 100 meters, considering the categorical and delicate nature of land use data?

We thank the reviewer for this observation. Regarding the fuel classes, we relied on the classification developed within the **FirEUrisk project**, designed for the pilot sites and able to differentiate broad vegetation types relevant to fire risk (e.g., coniferous forests, broadleaved forests, shrublands), although it does not reach the level of detail of specific species such as eucalyptus stands. For further details on the categories, we refer to Aragoneses et al. (2023) (https://essd.copernicus.org/articles/15/1287/2023/). The assignment table was provided in the manuscript as Supplementary Table S2. Concerning the projection from 10 m to 100 m, since this is a categorical variable, we applied a majority vote criterion, assigning each 100 m cell to the dominant class, which preserves spatial consistency. We have updated the supplementary material further details have been added in section 2.2.4 and the following table in supplementary material.

| FirEUrisk fuel type | | FirEUrisk fuel type | |
|---|---|---|---|
| Code | Description | Code | Description |
| 1111 | Open broadleaf evergreen forest | 23 | High shrubland [$\geq 1.5$ m] |
| 1112 | Closed broadleaf evergreen forest | 31 | Low grassland [0–0.3 m] |
| 1121 | Open broadleaf deciduous forest | 32 | Medium grassland [0.3–0.7 m] |
| 1122 | Closed broadleaf deciduous forest | 33 | High grassland [$\geq 0.7$ m] |
| 1211 | Open needleleaf evergreen forest | 41 | Herbaceous cropland |
| 1212 | Closed needleleaf evergreen forest | 42 | Woody cropland |
| 1221 | Open needleleaf deciduous forest | 51 | Wet and peat/semi-peat land – tree |
| 1222 | Closed needleleaf deciduous forest | 52 | Wet and peat/semi-peat land – shrubland |
| 1301 | Open mixed forest | 53 | Wet and peat/semi-peat land – grassland |
| 1302 | Closed mixed forest | 61 | Urban continuous fabric |
| 21 | Low shrubland [0–0.5 m] | 62 | Urban discontinuous fabric |
| 22 | Medium shrubland [0.5–1.5 m] | 7 | Nonfuel |

- **Line 208**: What exactly is meant by "null model"?

We thank the reviewer for this comment. By *"null model"* we were referring to an *initial model*, in which we optimized the hyperparameters of the Random Forest algorithm for each Pilot Site before running the final experiments. To avoid confusion, we have replaced the term *null model* with *initial model* throughout the manuscript (L214).

- **Line 210**: The term *number of predictors* should be highlighted, perhaps using italics or quotation marks, for clarity.

Done.

- **Line 214**: Reference to "Section 0" is likely a formatting or numbering error and should be corrected.

(Removed from text.

- **Line 214 (continued)**: Was the Autocorrelation Control (AC) also used in the full model with all the regions? If so, what was the bounding box adopted?

Yes, the Autocorrelation Control (AC) term was also used in the full model. The bounding box was defined by the extreme coordinates of the pilot sites: the

northernmost point of PS1 (Northern Europe), the easternmost point of PS5 (East Attica), and the southernmost and westernmost points of PS3 (Central Portugal). Please see section 2.4.1 (L223-L225).

- **Line 217**: The AC strategy deserves more clarification. It seems to include:

  - Distance from the center of each PS

  - Distance from the corners of each PS bounding box

  - x and y coordinates in the adopted CRS

Yes, the AC strategy includes these variables, which correspond to the Euclidean Distance Features (EDFs) described by Milà et al. 2024 (https://gmd.copernicus.org/articles/17/6007/2024/). These variables were incorporated to account for spatial autocorrelation effects. Please see 2.4.1 section (L220-223).

Given this, I would expect that a local PS model relying too heavily on the AC variables (as visible from importance rankings) could be overfitting the ignition patterns of its training set rather than capturing true statistical drivers of ignition. The dummy variable for the PS code used in the full model seems to act similarly to the AC, but at a larger scale. Could you please clarify what is meant by "The AC control successfully alleviated spatial autocorrelation…" (line 217) and confirm whether "disregarding AC control" means simply removing all AC variables from the feature set?

In our study, Euclidean Distance Features (EDF) were included to control for spatial autocorrelation. It is true that EDF can potentially induce overfitting (Milà et al., 2024), particularly when these features rank among the top predictors in variable-importance measures. Consequently, importance ranks in models that include such predictors should be interpreted with great caution, and emphasis should be placed instead on explanatory variables (Meyer et al., 2019; Wadoux et al., 2020). Please check Reviewer 1 author's reply, where we show the results of a model's overfitting performance. Otherwise, the same plot is included in supplementary material (Figure S1)

Regarding the clarification on line 217, it should be noted that incorporating the spatial autocorrelation (AC) term as a predictor substantially reduced the number of realizations showing residual spatial autocorrelation, with an overall reduction of about 66% across pilot sites.

Meyer, H., Reudenbach, C., Wöllauer, S., and Nauss, T.: Importance of spatial predictor variable selection in machine learning applications – Moving from data reproduction to spatial prediction, Ecol. Model., 411, 108815, https://doi.org/10.1016/j.ecolmodel.2019.108815, 2019.

Milà, C., Ludwig, M., Pebesma, E., Tonne, C., and Meyer, H.: Random forests with spatial proxies for environmental modelling: opportunities and pitfalls, Geosci. Model Dev., 17, 6007–6033, https://doi.org/10.5194/gmd-17-6007-2024, 2024 .

Wadoux, A. M. J.-C., Samuel-Rosa, A., Poggio, L., and Mulder, V. L.: A note on knowledge discovery and machine learning in digital soil mapping, Eur. J. Soil Sci., 71, 133–136, https://doi.org/10.1111/ejss.12909, 2020

- **Figure 2 (Line 250)**: This figure highlights an extremely important aspect. The choice of presence and pseudo-absence points can shift AUC from 0.4 to 0.9, as in the case of East Attica. This issue is critical and often neglected in wildfire susceptibility literature.

  This variability evidences the poor performance of the model in this pilot site due to data scarcity and uneven ignition distribution. Expanding the ignition sample—if governmental datasets become open—or complementing the analysis with similar regions could substantially improve model robustness in eastern mediterranean basin. To amend this limitation, we employed repeated subsampling and reported the distribution of AUC values rather than relying on a single realization. For cartographical representation we select the model that has an AUC closer to median AUC of the different repetitions. We now stated the East Attica overfitting problem in 3.1 (L 253-254) and 4.4 sections (L421-L423).

- **Line 260**: Please specify that "dry and warm season" refers to the Swedish climate, which may not be intuitively understood by all readers.

  We added this clarification: "*which in the context of the Northern Europe climate correspond to its relatively warm and dry summer months*" in lines 270-271

- **Line 266 and elsewhere**: The phrase "chances of ignition" may be misleading. Human sources of ignition (e.g., arson, negligence) are usually orders of magnitude higher than the fires actually recorded. What determines whether a fire is recorded is its success in developing beyond a minimal threshold. It would be more precise to refer to "chances of successful ignition" throughout the manuscript.

  Changes done.

- **Caption of Table 2**: The sentence should be revised to: "The top 5 variables **for each column** are highlighted in grey."

  Done,

- **Figure 4 (Line 310)**: While the figure is clear and well done, the discussion could be enriched by acknowledging a critical issue in model interpretation: susceptibility values from RF (ranging 0 to 1) cannot be meaningfully compared across pilot sites. A 0.999 value in Sweden does not equate to a 0.999 in Attica. This is where the full model provides value by smoothing across regions. In some of my previous work, I have addressed this using quantile ranking—i.e., describing a pixel as "top 5% susceptibility" within its region, rather than relying on the raw RF voting score. Consider discussing this approach or acknowledging the issue.

  We thank the reviewer for highlighting this important point. As noted, raw Random Forest (RF) probability values are calibrated from the local training data of each pilot site and thus cannot be directly compared across regions—a 0.999 value in Sweden does not represent the same absolute risk as a 0.999 in Attica. The full continental

model partly mitigates this issue by pooling data across sites, which smooths differences in scale. Importantly, our focus is not on comparing absolute probability values between sites, but rather on analyzing the drivers and spatial distribution of ignitions. We now explicitly refer to probability values in terms of quintiles in section 3.3 and updated the figure 4. in which the results are represented by quintiles. Following your suggestion, this approach improves interpretability and highlights the relative distribution of probabilities more clearly.

- **Line 325**: Since temporal variability is removed by taking average values, I assume your model highlights "spatial areas where extreme dry events tend to occur" rather than correlating specific years with ignition. Is that correct?

  That is correct in part. The mean values capture the general spatial tendency of where extreme dry events are more likely to occur, as in Mediterranean basin. However, by also incorporating Z-Scores, we account for interannual variability, allowing the model to identify whether a given year is characterized as extreme or not.

- **Line 339**: "Modest climate conditions" is ambiguous. I believe what you mean is that Attica's fire season is, under climatic/weather conditions, uniformly and persistently extreme. It would be more accurate to use the phrase "fire-prone climate" here instead of "favourable climate".

  Changed

**Additional Literature Suggestions**

- **On RF versus other techniques, and wildfire susceptibility modelling performance** (Section 2.4, line 195): Trucchia, A., Izadgoshasb, H., Isnardi, S., Fiorucci, P., Tonini, M. (2022). *Machine-Learning Applications in Geosciences: Comparison of Different Algorithms and Vegetation Classes' Importance Ranking in Wildfire Susceptibility*. Geosciences, 12(11), 424. https://doi.org/10.3390/geosciences12110424

  Incorporated in line 205

---

## Author Response (AR2)

**Response to reviewer - EGUSPHERE-2025-143**

**Lines 197–198: The text mentions ERA5-Land, but elsewhere the work appears to use ERA5 (coarser native resolution). Please confirm which product(s) were used and ensure consistency throughout. To improve reproducibility, explicitly list for each dataset: native spatial/temporal resolution, temporal coverage, variables used, and all post-processing steps (e.g., temporal aggregation, resampling, regridding). A compact table would be acceptable for that and would improve the manuscript.**

*We thank the reviewer for pointing this out. We confirm that ERA5-Land was used for this study at its native 0.1° spatial resolution. The manuscript has been updated including the correct citation for ERA5-Land and Table S1.*

*Following the reviewer's recommendation, Table S1 has been updated to include the following additional information for each dataset: native spatial resolution/scale and post-processing applied.*

**Line 200: Please revise the climatological baseline to align with WMO practice: 1991–2020 (not 1991–2021). Adjust wording and any affected calculations/figures accordingly.**

*We thank the reviewer for the suggestion regarding the climatological baseline. While it is true that the WMO reference period is 1991–2020 should be used, we have decided to retain the 1991–2021 baseline for this study because ignition samples from 2021 are available for PS2 (Central Europe) and PS5 (East Attica), and which is crucial in PS5, as it is the area with the fewest records.*

*To verify that this choice does not affect the results, we quantified the absolute differences between the original (1991–2021) and WMO (1991–2020) baseline rasters, aggregated across the full spatial and temporal extent of each pilot site. The resulting mean absolute differences are very small (PS1: 0.038; PS2: 0.027; PS3: 0.114; PS4: 0.038; PS5: 0.147), ensuring that retaining the 2021 data introduces only minor differtences and does not affect the conclusions of the study.*

**Line 108: There is a visible error message (likely a broken cross-reference). Please fix the cross-reference and scan the manuscript (including captions and supplement references) to ensure no similar messages remain.**

*Checked*

**Line 254: Typo. Correct to "Southern Sweden"**

*Done*

**Line 222: Typo. Correct to "Euclidean distance"**

*Done*